# Analysis of Control Characteristics between Dominant and Non-Dominant Hands by Transient Responses of Circular Tracking Movements in 3D Virtual Reality Space

**DOI:** 10.3390/s20123477

**Published:** 2020-06-19

**Authors:** Wookhyun Park, Woong Choi, Hanjin Jo, Geonhui Lee, Jaehyo Kim

**Affiliations:** 1Department of Mechanical and Control Engineering, Handong Global University, Pohang 37554, Korea; 21834002@handong.edu (W.P.); 21834006@handong.edu (H.J.); 21834004@handong.edu (G.L.); 2Department of Information and Computer Engineering, National Institute of Technology, Gunma College, Maebashi 371-8530, Japan

**Keywords:** circular tracking movement, motor control, virtual reality, dominant hand, transient response

## Abstract

Human movement is a controlled result of the sensory-motor system, and the motor control mechanism has been studied through diverse movements. The present study examined control characteristics of dominant and non-dominant hands by analyzing the transient responses of circular tracking movements in 3D virtual reality space. A visual target rotated in a circular trajectory at four different speeds, and 29 participants tracked the target with their hands. The position of each subject’s hand was measured, and the following three parameters were investigated: normalized initial peak velocity (*IPV2*), initial peak time (*IPT2*), and time delay (*TD2*). The *IPV2* of both hands decreased as target speed increased. The results of *IPT2* revealed that the dominant hand reached its peak velocity 0.0423 s earlier than the non-dominant hand, regardless of target speed. The *TD2* of the hands diminished by 0.0218 s on average as target speed increased, but the dominant hand statistically revealed a 0.0417-s shorter *TD2* than the non-dominant hand. Velocity-control performances from the *IPV2* and *IPT2* suggested that an identical internal model controls movement in both hands, whereas the dominant hand is likely more experienced than the non-dominant hand in reacting to neural commands, resulting in better reactivity in the movement task.

## 1. Introduction

Researchers in various academic areas have studied human movement to better understand human motor control mechanisms [1,2,3,4,5,6,7,8,9,10,11,12,13,14,15,16,17,18,19,20,21,22,23,24,25]. In particular, the human body produces a single movement through the combined coordination of multiple joints and muscles [18,26], while the human brain controls physiological movements by receiving sensory feedback and transmitting neural commands. As such, many individual movements involve a variety of motor control behaviors [19,20,21,22,23,24,25,26,27,28,29,30,31,32,33,34,35,36,37,38,39,40,41,42,43,44,45,46,47,48].

Various target-tracking movements have been studied using a multi-joint upper-limb to analyze human visuo-motor systems [26,27,28,29,30,31,32,33,34,35,36,37,38,39,40,41,42,43,44,45,46,47,48]. Several studies have used a visual target moving in a straight line [44,45,46,47] or a sinusoidal waveform [26,27,28,29] to examine motor characteristics on a 1-DOF arm movement, and many have used target-tracking on a circular trajectory to investigate the velocity-dependent behaviors of upper-limb movement [33,34,35,36,37,38,39]. The control characteristics of the visuo-motor system have also been examined to ascertain their contribution to intermittency [28,29,30,31,32] and proactiveness in response to a predictable target [26,39,40,41].

Handedness and hand usage are another important aspect of upper-limb movement [47,48,49,50,51,52,53]. Several efforts have been made in the physiological field to elucidate which factors determine the preferred or dominant hand (*DH*) [52,53]. Functional comparisons between the *DH* and non-*DH* (*NDH*), as well as between right- and left-handed individuals, have investigated the control characteristics of upper-extremity movement [47,48,49,50,51]. These studies revealed that control performance differed between the hands when the given movement task required a high level of accuracy or dexterity [47,48], although handedness does not directly cause the performance differences between the hands [49,50].

Recently, we implemented a system that measures the Cartesian movement of an end-effector within a three-dimensional virtual reality environment, enabling quantitative evaluation of human hand movement during various movements, such as reaching and tracking [37,38]. In one study, we examined a visual target rotated along a specific circular trajectory, as well as subjects’ tracking performances under monocular and binocular vision [37]. In another, we analyzed how the human visuo-motor system controls temporo-spatial parameters during circular tracking movements [38].

Circular target-tracking tasks have been distinguished from other movement tasks because the continuous motion allows researchers to analyze the velocity properties of upper-limb end-point movement [25,32,33]. Previous studies have compared subjects’ control of stable behaviors at different target speeds during circular tracking tasks [37,38].

To comprehend the control mechanisms of the human upper-extremity, researchers must examine the initial behaviors in a tracking movement. In control engineering, the time-domain response of a controlled system is classified into a transient response and a steady-state response [54]. For instance, assuming a step signal is given to a system, the rise time and overshoot are frequently evaluated in the performance analysis of the system’s transient response. Steady-state error is often determined to examine the stability of a control system’s steady-state response. Considering the human visuo-motor system as a controlled system, the transient responses of the target-tracking movement under various target speeds are thought to indicate the velocity-dependent motor control mechanism.

Therefore, in the present study, velocity-dependent motor control characteristics were examined using the transient responses of the circular tracking movements of the *DH* and *NDH* in a 3D space. Temporal behaviors in response to a moving target were analyzed using three parameters: the normalized magnitude of the hand’s initial peak velocity on the 2D plane (*IPV2*), the corresponding initial-peak time (*IPT2*), and the calculated time delay of the hand (*TD2*). Participants used their *DH* and *NDH* to track a visual target rotating in a circular trajectory at four different speeds.

## 2. Materials and Methods

### 2.1. Subjects and Experimental Setup

The present study involved 29 subjects: 19 males and 10 females (Table 1). Their mean age was 24.67 ± 3.1 years, and 26 of them were right-handed while the others were left-handed. All subjects had normal or corrected-to-normal vision and decent physical condition without any medical history regarding their upper limbs. The subjects had no experience of similar studies or any relevant measurement devices utilized in this experiment, and all of them provided written consent prior to participation in the research. All experiments were performed in compliance with relevant guidelines and regulations. The protocol was approved by the ethics committee of the National Institute of Technology, Gunma College.

The experiment aimed to investigate visuo-motor control characteristics at different target speeds at the beginning of visual guided tracking tasks using a virtual-reality (VR) system. Such VR systems must (1) provide an immersive 3D audio-visual environment, (2) visualize preprogramed 3D objects and operate them in real-time, and (3) record and display the movement of the user’s hand in the environment [37]. HTC Vive was used to give visually guided tracking tasks to the subjects and to display the position of their hands in real time. The head-mounted display (HMD) offered 2160 × 1200-pixel graphic resolution at a refresh rate of 90 Hz. The compatible controller weighed 200 g and was virtually illustrated as 20 cm long with a white stick in the VR space.

An experimental chair designed for the research was placed in the center of the camera-viewed space. The subjects sat on this chair, and two belts fixed on each side were used to fasten their upper bodies to prevent undesired shoulder movements.

### 2.2. Experimental Procedures

The subjects were fully informed about the research and safely prepared for the experiment. To start with, they were instructed to read the written notice thoroughly and given verbal instructions by the experimenter. They were then asked to fill in a consent form and assigned to the experimental chair. After they were seated, they were carefully fastened onto the chair and received a VR face-protection cover. They were then equipped with the HMD and grasped a hand-held controller (Figure 1A). To become acquainted with and to practice interaction within the provided VR environment, individual subjects were assigned 20 min of a familiarization session with basic VR programs embedded in the HTC Vive system.

Once the subjects sufficiently performed the session, a brief checkup was given to verify if the subjects find any inconvenience prior to the experiment, and the experimental program was initiated. The HMD showed a target with a red, ball-shaped sphere and a tracer with a yellow, ball-shaped sphere that had a white stick attached to it. Both had a radius of 15 mm but differed in size depending on their depths (Figure 1B). The background was black (Figure 1C).

The controller was custom calibrated to the arms of each individual subject. Specifically, the subject initially placed the controller immediately beneath their jaw and pulled the trigger. The experimenter then told the subject to reach their arm straight forward and release the trigger after full reaching. The tracer was continuously displayed on the HMD, and the target appeared after the trigger was released. The target was set at a position of 200 mm in the positive direction on the *Z*-axis and 150 mm in the positive direction on the *Y*-axis from the position of the tracer. This process was executed when the subject changed their hand to grab the controller.

The subjects were asked to place the tracer in the center of the target to verify whether the calibration was appropriate. They were then given verbal corrections to ensure they were aware of the initial position of the target. No physical adjustment was applied to the subjects.

The subjects were then directed to maintain the position of the tracer. They were also informed that four “beep” sounds would be played to indicate a 3-s countdown and that the subjects should start to track the target as closely as possible using the tracer after the 4th beep sound, regardless of the target speed. When they accurately placed the tracer and maintained their posture, the experimenter notified the start of a task and ran the program. The speaker on the HMD produced the beep sounds, and the subject started tracking immediately after hearing the last sound.

### 2.3. Movement Tasks

The experiment consisted of two main tasks. The first was to track the target with the tracer using the *DH*. The target, displayed at the initially calibrated position (0, 150, 0 mm), moved in a circular trajectory in the clockwise direction on the frontal (*X*-*Y*) plane. The radius of the orbit was 150 mm, and the target revolved six times at a constant speed. The target speeds were set at 0.250 π, 0.500 π, 1.000 π, and 1.500 π rad/s along the trajectory. Only one trial was carried out at each speed, and the subjects were told that the target would move faster in the next trial. A break of 30 s was given after each trial.

The second task was to perform the same type of tracking movement using the *NDH*. At the beginning of the experiment, each subject was randomly assigned which hand would start the tracking movement: *DH* or *NDH*. The controller was re-calibrated for each hand.

### 2.4. Data Analysis

Real-time positions of the tracer in the Cartesian coordinates (*X*,*Y*,*Z*) were used for data analysis. The sampling frequency was 90 Hz and the transient response of the circular tracking motion was examined using the data measured after the target started moving. As a transient response of a controlled system illustrates behavior to instantaneous changes in its reference input and a process of reaching a steady-state response, so the initial performance of the tracer indicated a transient response of how the subject instantly started the target-tracking movement.

Three parameters were chosen to analyze the performance of the transient response. The first was the *IPV2*, the initial peak velocity of the tracer on the frontal plane. The measured *X* and *Y* values were numerically differentiated and computed with respect to the speed of the tracer. Considering the target speed as a referenced input, the *IPV2* could be interpreted as an overshoot of the transient response in a controlled system. The *IPV2* was normalized to the corresponding target speed to allow comparisons with other *IPV2* values at different target speeds. The *IPV2* was calculated as a percentage.
(1)$$v_{x,trc}\left[t\right]=\left\{\begin{array}{cc} \frac{x_{trc}\left[t\right]-x_{trc}\left[t-1\right]}{T_{s}} & \mathrm{for}\;t>1\\ 0 & \mathrm{for}\;t=1 \end{array}\right.$$
(2)$$v_{y,trc}\left[t\right]=\left\{\begin{array}{cc} \frac{y_{trc}\left[t\right]-y_{trc}\left[t-1\right]}{T_{s}} & \mathrm{for}\;t>1\\ 0 & \mathrm{for}\;t=1 \end{array}\right.$$
(3)$$v_{xy,trc}\left[t\right]=\sqrt{v_{x,trc}{\left[t\right]}^{2}+v_{y,trc}{\left[t\right]}^{2}}$$
(4)$$v_{xy,trc}^{\mathrm{peak}}={\left.\max\left(v_{xy,trc}\left[t\right]\right)\right|}_{t=1}^{n_{hr}}$$
(5)$$v_{xy,trg}=r_{traj}\times \omega_{trg}$$
(6)$$\mathrm{IPV}2=\frac{v_{xy,trc}^{\mathrm{peak}}}{v_{xy,trg}}\times 100$$

Constant *n_hr_* denotes the length of data measured at a sample rate of 90 Hz while the target rotated through half of the first orbit. The *n_hr_* values at each speed were 360 (0.250 π rad/s), 180 (0.500 π rad/s), 90 (1.000 π rad/s), and 60 (1.500 π rad/s). The abbreviations *trg* and *trc* represent the target and tracer, respectively. The velocity of the target on the frontal plane, *v_xy,trg_*, was decided by the target’s angular speed *ω_trg_* which was one of 0.250 π, 0.500 π, 1.000 π, and 1.500 π rad/s. The radius of the target’s trajectory $r_{\mathrm{traj}}$ was fixed to 150 mm and the *v_xy,trg_* was considered as a constant because the target rotated the circular trajectory at a constant velocity during a single trial.

The second parameter was *IPT2*, the initial peak time, which is the time at which *IPV2* occurred.
(7)$$\mathrm{IPT}2={\left.t\left[v_{xy,trc}^{peak}\right]\right|}_{t=1}^{n}$$

The last parameter was *TD2*, the time delay or reaction time of the tracer on the frontal plane. In order to examine the transient response, it is essential to know the starting moment of the response. Other studies could easily distinguish the moment of when the subject actually started to move since they employed measurement devices with 1- or 2-DOF [26,27,28,29,30,31,32,33,34,35,36,37,38,39,40,41,42,43,44,45,46,47,48]. Such devices were fixed on a certain spot or plane, and the measured data did not fluctuate for a stationary movement, allowing easy detection of positional changes of the tracer. The VR environment of this experiment, however, did not physically restrict any movement of our subjects. In fact, they placed their hands in the air without any structural support. Though the subjects were told to stay still, physiological factors such as cardiac impulses or a hand tremor may cause positional fluctuation of the hands. These unintentional movements hinder the detection of the starting moment of the response. By identifying features of the involuntary movements, the transient response of the target tracking movement can be discriminated from the unintentional fluctuation.

In statistics, one standard deviation of a normal distribution denotes how much difference each data point has from its mean value, expressing the broadness of the data. In a normal distribution, 68.2% of the data lies within one standard deviation of the mean, and three standard deviations cover 99.6% of the sample.

Applying this theory to our experiment, one standard deviation of the tracer during a 3-s countdown (*STDtrc*) represented a statistic feature of the involuntary movements prior to the tracking movement. That is, *STDtrc* could be referred to as a magnitude of the hand’s tremor or how widely the tracer was fluctuated irrespective of the subject’s intention. If the displacement of the tracer from its initial position was greater than three standard deviations, this could be interpreted as the subject moves the tracer with clear intention. The *TD2* value expressed the moment at which the tracer’s displacement from its initial position became more than three times that of the *STDtrc*.
(8)$${\mathrm{disp}}_{trc}\left[t\right]=\sqrt{{\left(x_{trc}\left[t\right]-0\right)}^{2}+{\left(y_{trc}\left[t\right]-150\right)}^{2}}\;\mathrm{for}\;-270\leq t\leq -1$$
(9)$${\mathrm{disp}}_{trc}^{mean}=\frac{\sum_{t=-270}^{-1}{\mathrm{disp}}_{trc}\left[t\right]}{n}$$
(10)$${\mathrm{STD}}_{trc}=\sqrt{\frac{\sum_{t=-270}^{-1}{\left({\mathrm{disp}}_{trc}\left[t\right]-{\mathrm{disp}}_{trc}^{mean}\right)}^{2}}{n-1}}$$
(11)$$d_{xy,trc}\left[t\right]\;\left[\mathrm{mm}\right]=\sqrt{{\left(x_{trc}\left(t\right)-x_{trc}^{init}\right)}^{2}+{\left(y_{trc}\left(t\right)-y_{trc}^{init}\right)}^{2}}\;\mathrm{for}\;t>0$$
(12)$$\mathrm{TD}2={\left.\min\left(t\left[d_{xy,trc}\left(t\right)\geq 3\times {\mathrm{STD}}_{trc}\right]\right)\right|}_{t=1}^{n_{hr}}$$

$disp_{trc}$ represented a displacement of the tracer from the target’s position (0, 150, 0 mm) on the frontal plane. Since the sampling frequency was 90 Hz, 270 position data were measured during the 3-s countdown (the time indices ranged from −270 to −1 to distinguish from the data measured after the countdown). Equation (8) indicated that displacements were calculated for the three seconds before the start. ${\mathrm{disp}}_{trc}^{mean}$ in Equation (9) is a mean value of the collected ${\mathrm{disp}}_{trc}$, which is used for computing the value of *STDtrc* in Equation (10). After the 4th beep, the subject may initiate to move the tracer, but the initial position of the tracer may differ from that of the target, although the subjects were told to place the tracer in the center of the target prior to the experiment. The tracer is on the frontal plane ($x_{trc}^{init}$, $y_{trc}^{init}$) at the start of each trial. Displacements after the start were calculated as $d_{xy,trc}$ as shown in Equation (11), and the moment that the displacement becomes greater than triple of *STDtrc* was considered as *TD2*.

All these parameters were calculated from data measured for two types of hand factors: *DH* and *NDH*, as well as four types of speed factors: *S1*, 0.250 π rad/s; *S2*, 0.500 π rad/s; *S3*, 1.000 π rad/s; *S4*: 1.500 π rad/s. Through examinations of each parameter, hand and speed-dependent motor characteristics are to be studied. To be specific, the control mechanism of how humans respond to an instantaneous velocity input is to be investigated. Through *IPV2*, a magnitude-related property in velocity-control process and accuracy of velocity prediction can be examined while *IPT2* and *TD2* directly propose temporal behaviors of velocity-control performance such as reactivity or sensitivity to a given velocity input.

Performances of the 29 subjects were analyzed using MATLAB 2019a, MathWorks (Natick, Massachusetts, United States). A Butterworth IIR digital filter (3rd-ordered low-pass with a cutoff frequency of 7.5 Hz) was applied to all the data prior to analysis. SPSS statistics V26 (IBM) was used to carry out two-way repeated-measures analysis of variance (ANOVA) tests to numerically analyze differences in performances of the circular tracking movements with the hand and speed factors. Mauchly’s sphericity test (*p* > 0.05) was also executed to validate the results of ANOVA. For the results whose sphericity was not assumed (*p* < 0.05), the values corrected with Greenhouse–Geisser in tests of within-subjects effects were employed. The post hoc test was performed by Bonferroni pairwise comparison (*p* < 0.05) to verify the significance of each condition for hand and speed factors.

## 3. Results

The designed experiment was to investigate initial motor characteristics in the target tracking movements. To examine speed-dependent characteristics of the transient responses and the differences between the *DH* and *NDH*, 29 subjects participated in the experiment using their *DH* and *NDH* at the four speeds. The obtained tracer’s position data was transformed into three parameters: *IPV2*, *IPT2*, and *TD2*.

Figure 2 shows typical examples of the calculated velocities of target-tracking movements at *S2*, while the target rotated through the first half-orbit of the circular trajectory. The measured trajectories of the *DH* and *NDH* are illustrated on the frontal plane (Figure 2A,B). The velocity components of the hands are depicted on the *X*-axis (Figure 2C1,D1) and *Y*-axis (Figure 2C2,D2). The normalized speeds are displayed in Figure 2C3,D3 in terms of the target speed *S2*; the computed *IPV2*, *IPT2*, and *TD2* are marked.

### 3.1. Differences in the Transient Response of Tracking Movement Based on IPV2

ANOVA of the *IPV2* demonstrated that only the speed factor caused a significant difference in the *IPV2* of the transient responses of the circular tracking movement (item A in Appendix A). The hand factor and the interaction between the hand and speed factors did not significantly affect the *IPV2*.

Figure 3 illustrates the *IPV2* results of the *DH* and *NDH* in terms of the transient responses of the tracking movement on the 2D frontal plane. The statistical values of *IPV2* are listed as the variable *IPV2* in Table 2. There was no significant difference in *IPV2* between the *DH* and *NDH* (Figure 3A,B; item B in Appendix A). The *IPV2* of the *DH* indicated that the maximum speed differences between the target and the tracer were approximately 37.70% and 26.22% larger at *S1* and *S2*, respectively, than at *S4* (Figure 3C; item C in Appendix A). The *IPV2* of the *NDH* indicated that the maximum speed differences between the target and the tracer were approximately 55.05% and 35.30% larger, respectively, at *S1* and *S2* than at *S3* as well as 62.79% and 43.04% larger at *S1* and *S2* than at *S4* (Figure 3D; item D in Appendix A). The combined *IPV2* suggested that the maximum speed differences between the target and the tracer were approximately 39.25% and 23.63% larger at *S1* and *S2* than at *S3* as well as 50.25% and 34.63% larger at *S1* and *S2* than at *S4* (Figure 3E; item E in Appendix A).

These results signified that *IPV2* was likely to decrease as the target speed became faster on the 2D frontal plane regardless of the hand. In other words, the subjects’ hands showed clear tendencies to reduce the initial speed difference by approximately 16.75% at a faster speed.

### 3.2. Differences in Transient Response of Tracking Movement Based on IPT2

ANOVA of the *IPT2* revealed that only the hand factor caused significant differences in the *IPT2* of the transient responses of the circular tracking movement (item A in Appendix A). The speed factor and the interaction between the hand and speed factors did not significantly affect the *IPT2*.

Figure 4 illustrates the *IPT2* results of the *DH* and *NDH* in terms of the transient responses of the tracking movement on the 2D frontal plane. The statistical values of *IPT2* are listed as variable *IPT2* in Table 2. Although no significant differences were found in the paired *IPT2* (Figure 4A; item B in Appendix A), the *IPT2* of the hands revealed that the *DH* reached the initial peak speed approximately 0.0423 s before the *NDH* did, regardless of the target speed (Figure 4B). The *IPT2* values of the *DH* and *NDH* indicated that there were no significant differences in the speed factor (Figure 4C–E; items C–E in Appendix A).

The stated results implied that the *IPT2* had a significant relationship with the hand factor on the 2D frontal plane. Specifically, the subjects’ *DH* attained its initial peak velocity 0.0423 s before the *NDH* did at the beginning of the tracking movement while both the hands disclosed no clear tendency on the target speed.

### 3.3. Differences in Transient Response of Tracking Movement Based on TD2

ANOVA of the *TD2* indicated that all the factors and their interaction caused significant differences in the *TD2* of the transient responses of the circular tracking movement (item A in Appendix A).

Figure 5 illustrates the *TD2* results of the *DH* and *NDH* during the transient responses of the tracking movement on the 2D frontal plane. The statistical values of *TD2* are listed as variable *TD2* in Table 2. Paired analysis of *TD2* implied that the subjects initiated tracking approximately 0.0494 s earlier at *S1* when using the *DH* than when using the *NDH*; at *S2*, they initiated tracking 0.0709 s earlier (Figure 5A; item B in Appendix A). The *TD2* of the hands revealed that the *DH* started moving the tracer approximately 0.0417 s earlier than the *NDH*, irrespective of the target speed (Figure 5B).

The *TD2* of the *DH* indicated that the subjects initiated tracking approximately 0.0406 s earlier at *S2* than at *S1*, and 0.0567 s earlier at *S4* than at *S1* (Figure 5C; item C in Appendix A), whereas the *TD2* of the *NDH* indicated that the subjects reacted to the target approximately 0.0716 and 0.0743 s earlier at *S3* and *S4* than at *S1*, and that they reacted 0.0525 and 0.0552 s earlier at *S3* and *S4* than at *S2* (Figure 5D; item D in Appendix A). The combined *TD2* suggested that the subjects started moving the tracer approximately 0.0299, 0.0544, and 0.0655 s earlier at *S2*, *S3*, and *S4*, respectively, than at *S1*, as well as 0.0356 s earlier at *S2* than at *S4* (Figure 5E; item E in Appendix A).

These results signified that *TD2* showed significant differences between slow (*S1*, *S2*) and fast (*S3*, *S4*) speeds. In other words, the subjects reacted to the target approximately 0.0218 s more quickly at fast target speeds in the circular tracking movement. Moreover, the *TD2* of the *DH* and *NDH* showed different statistical features; specifically, the subjects had a transient response that was approximately 0.0417 s earlier with the *DH* than with the *NDH*.

## 4. Discussion

This study quantitatively analyzed the transient responses of the circular tracking movement of the *DH* and *NDH* to investigate initial motor control characteristics at different target speeds. Three parameters were examined: *IPV2*, the normalized magnitude of initial peak velocity, *IPT2*, initial peak velocity time, and *TD2*, the reaction time the subject took to react to the target. All parameters were investigated at *S1*–*S4* with the *DH* and *NDH*.

The following discussion mainly focuses on: (1) speed-dependent features of *IPV2*, (2) temporal analysis of *IPT2* and *TD2*, and (3) clinical applications and future work.

### 4.1. Speed-Dependent Features of IPV2

The subjects performed the target-tracking movement by immediately moving the tracer faster than the target after they perceived target movement. As overshoot is a key component of transient response analysis in control theories [54], the speed of the tracer exceeded that of the target and, after the target was caught up, the subjects continued the tracking task by adjusting the speed of the tracer (Figure 2). Particularly at *S4*, the subjects controlled the speed of the tracer rather than reducing the position error between the target and the tracer. This implied that they deemed the target too fast to track and that they simply decided to maintain a certain level of the tracer’s speed.

Analyzing the immediate movements of the tracer when the target started moving, we showed that the *IPV2* decreased as the target speed increased, indicating that the initial prediction about the target speed was more accurate at faster speeds. The subjects could barely apprehend the next target speed because the given instruction was not quantitative, so the initial target-tracking movement was executed based solely on the subject’s predicted speed. The *DH* and *NDH* commonly had lower *IPV2* values at faster speeds and showed nearly identical behavior in overall speed, indicating that both hands have the same internal model for initiating tracking movement.

### 4.2. Temporal Analysis of IPT2 and TD2

The *IPT2* represents the time corresponding to *IPV2* and therefore indicates the moment when the subject first corrected the speed of the tracer toward the moving target. In other words, *IPT2* expresses the actual initiation of speed control. The *IPT2* showed no significant difference among the different speeds, but had a 0.0423-s difference between the *DH* and *NDH*. It follows that both hands may have nearly the same reactivity to the target speed. This also suggests that an identical internal model controls motor performances of the hands. Such an idea coincides with motor equivalence, the theoretical concept that the central nervous system (CNS) rather stores abstract forms of information on actions than records neural commands to specific muscles for performing actions [55]. The possibility for the hands to receive not distinct commands from the CNS but similar forms of information intimates that the *IPT2* could be a personal motor characteristic rather than an externally affected trait.

*TD2*, the time when the subject perceived the target’s positional change and initiated tracer movement, showed significant differences between the hands and among the various speeds. In the *DH* and *NDH*, the *TD2* reduced as the target became faster, showing clear differences in average values. Specifically, mutual differences were found between the slow (*S1*, *S2*) and fast (*S3*, *S4*) speeds, indicating that when the subjects performed more trials, they predicted the starting moment of trials more accurately. In addition, the *DH* had less *TD2* than the *NDH* at all the speeds, showing that the subjects were more proficient at controlling the *DH* than the *NDH*. Otherwise stated, the *DH*, comparatively the more experienced hand, reacted to the target faster than the *NDH*. Assuming that the same internal model controlled the movements of the *DH* and *NDH*, the tracers might have started moving at a nearly identical moment in each case. Since the *TD2* showed a significant difference between the hands, we concluded that the *DH* showed better performance at controlling objects in reacting to motor commands and catching the moving target.

### 4.3. Clinical Application and Future Work

Assessing of motor functions and analysis of possible recovery processes will be feasible by applying the protocol of this study to patients with restrictions on upper limb movements. The experimental system can measure performance of a circular target tracking movement and can evaluate motor characteristics of an upper limb quantitatively. The examined results can be integrated with conventional indicators which inspect motor capability of individual joint movements and diagnose current recovery processes from deficits of a neurological disorder, such as Fugl-Meyer assessment (FMA) and manual function test (MFT). After appropriate correlation analysis, the present protocol will be employed in estimating the patients’ physical conditions, for example, spasticity of their impaired limbs, and in speculating potential recovery phases.

The results of this research can benefit the clinical population with neural impairment by providing valuable information which can be employed in navigating suitable treatment. Such patients have difficulties in performing continuous multi-joint movements for a long time because of sensorimotor malfunction caused by the impairment. Motor characteristics of the patients can be examined by simply conducting a half orbit of the circular tracking movement. The obtained information can be effectively useful for therapists in planning proper rehabilitation goals or in designing customized treatment programs for the patients. The outcome of this research is expected to provide clinical improvement to those whose motor functions are deteriorated by a stroke, spatial neglect, or lesion in the cerebellum.

In future, we will investigate target-tracking performance at random target speeds without notifying the subjects about the target speed. Without any clues about the target speed, it will likely be difficult for subjects to make any predictions. The motor control characteristics of individuals will be studied by analyzing reactivity to a moving target.

*TD2* represented time delay or reaction time of the target tracking movement in this research. In neuroscience and biological engineering, reaction time is defined as the time taken when a neural command from the brain is transmitted to its musculoskeletal system for a joint movement, and the time is estimated as approximately 200 msec [45,46]. A time delay in control theories is normally referred to as a time necessary to reach 50% of the final value in a step response of a controlled system [54]. Both definitions were difficult to apply in this research since the measured data did not clearly indicate the starting moment of the reaction and final values of the subjects’ angular velocities in their tracking movement. We will evaluate these definitions by executing the same tasks within a dynamically-limited environment in future work.

A mathematical model to explain the tradeoff characteristic between speed and accuracy could be developed in the future. Our previous studies verified that the subjects could stably perform the tracking tasks after the first orbit of the trajectory [37,38]. In terms of control strategies, the subjects employed relatively more intermittent feedback control at the slow target speeds whereas feedforward control was dominant at the fast target speeds. This tradeoff relationship has been interpreted between speed and accuracy by establishing computational models [56,57]. Applying those mathematical models in our results is anticipated to better explain the speed-dependent characteristics of the human motor control mechanism.

Furthermore, an identical experiment with different weights of the hand-held controller will be conducted to investigate the effects of weight on the circular tracking movement. Applying a constant weight to the gravitational direction will probably influence the transient response of the tracking movement. Such an experiment should elucidate any differences in the impedance characteristics of *DH* and *NDH*, and analyze the motor characteristics of the hands from various perspectives.

## 5. Conclusions

The initial motor control characteristics of *DH* and *NDH* in the circular tracking movement were examined in a 3D VR space. The transient responses of the hands in the tracking movement were analyzed for *S1*–*S4* on the frontal plane. The *IPV2* decreased as the target speed increased, irrespective of the hand, suggesting that the subjects improved velocity-prediction accuracy by approximately 16.75% at faster speeds. The *IPT2* did not vary with the speed factor but showed a 0.0423-s difference between the *DH* and *NDH*, indicating that the *DH* reached its maximum speed earlier than the *NDH*. The *TD2* revealed significant differences in both the hand and speed factors, implying that the subjects reacted to the target 0.0218 s earlier as the target speed increased, and that the *DH* reacted to the target 0.0417 s earlier than the *NDH*. The performances of the *IPV2* and *IPT2* suggested that the same internal model controls tracking movements of both the hands whereas the results of *TD2* proposed that the *DH* generally had better reactivity in its transient response. The target tracking movements can be utilized in clinic environments to evaluate motor functions of patients and to set up treatment plans for rehabilitation. Effects of different experimental factors such as random speeds or weights will be analyzed as well as applying different methods of calculating reaction time and mathematical models of the tradeoff characteristics to the target tracking movements. This study examined the motor control characteristics of the *DH* and *NDH* in their transient responses of the circular tracking movements in the 3D VR space.

## Figures and Tables

**Figure 1 sensors-20-03477-f001:**
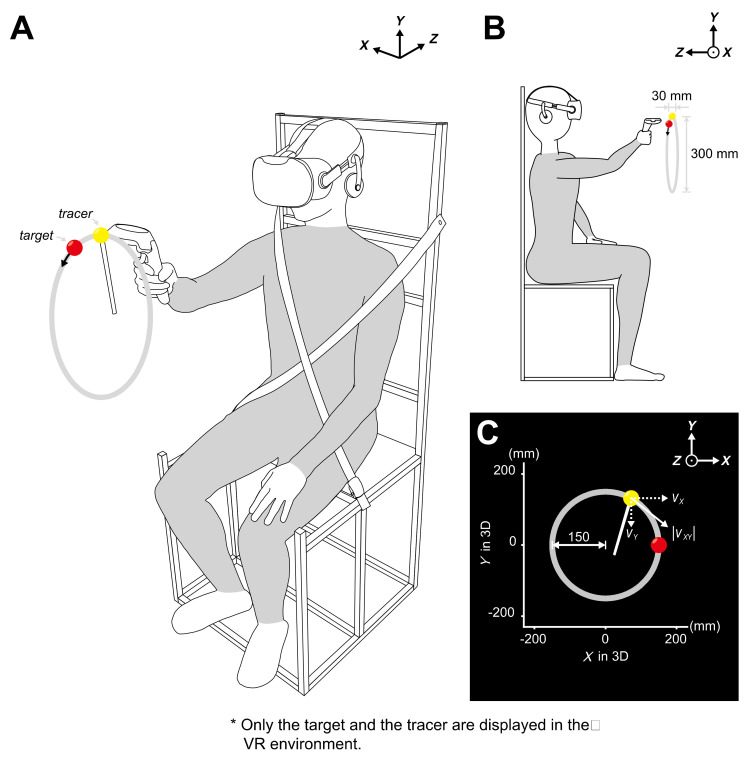
Experimental procedure for the circular tracking experiment. (**A**) Subject sitting on an experimental chair and holding a hand-held controller to conduct the target tracking movement with the virtual reality (VR) head-mounted display on. To restrict upper-body interference, belts are crossed on the subject’s chest. The target (red ball) is only visible in the 3D VR environment, while the tracer (yellow ball) indicates the current 3D location of the controller. The orbit of the target is invisible to the subject. (**B**) The sizes and trajectory of the balls. The target must move along a virtual trajectory with a diameter of 300 mm. They are instructed to follow the target with the tracer. (**C**) The target moves within the frontal plane and the velocity of the tracer is examined.

**Figure 2 sensors-20-03477-f002:**
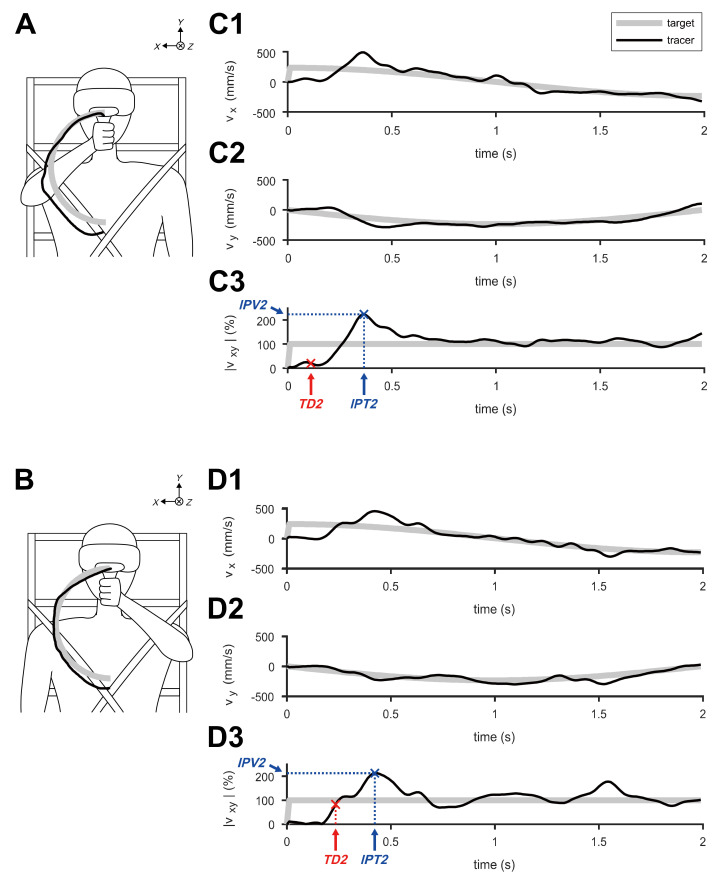
Typical examples of velocities during the first half-orbit of circular tracking movements at 0.500 π rad/s (S2). (**A**) Trajectories of the target and tracer performed by the dominant hand (*DH*) on the frontal plane. (**B**) Trajectories of the target and tracer performed by the non-dominant hand (*NDH*) on the frontal plane. (C1) Velocity of the *DH* calculated on the *X*-axis and (C2) on the *Y*-axis. (C3) Computed speeds on the frontal plane, normalized to *S2*. (D1) Velocity of the *NDH* calculated on the *X*-axis and (D2) on the *Y*-axis. (D3) Computed speeds on the frontal plane, normalized to *S2*.

**Figure 3 sensors-20-03477-f003:**
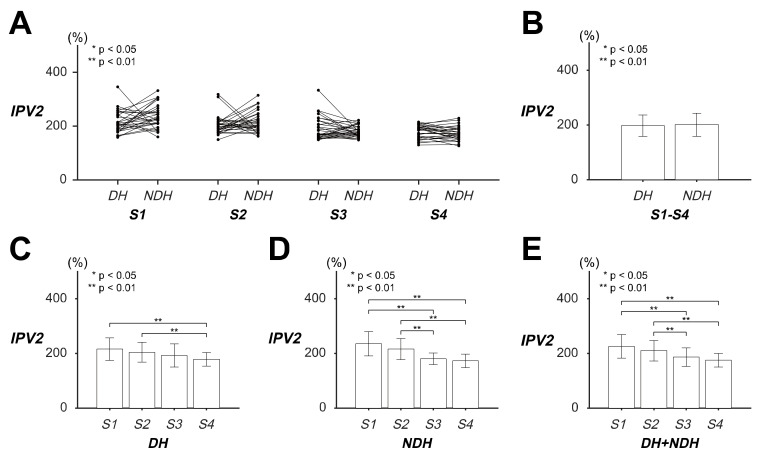
Evaluation of transient responses of the circular tracking movement based on initial peak velocity (*IPV2)* on the frontal plane. (**A**) The values of *IPV2* of 29 subjects for four speed factor levels. (**B**) The pairwise comparison between the dominant hand (*DH*) and *NDH* was shown for *IPV2* of all the speed factors. (**C**) Pairwise comparisons of *IPV2* were represented for the speed factor within *DH* and (**D**) within *NDH*. (**E**) The results of pairwise comparison of *IPV2* in both *DH* and *NDH*.

**Figure 4 sensors-20-03477-f004:**
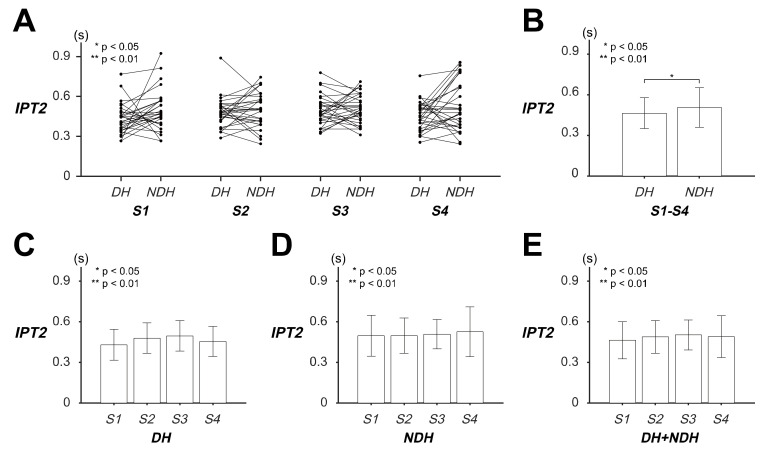
Evaluation of the transient responses of the circular tracking movement based on the initial peak time (*IPT2)* on the frontal plane. (**A**) The values of *IPT2* of the 29 subjects at four speed factor levels. (**B**) The pairwise comparison between dominant hand (*DH*) and non-dominant hand (*NDH*) is shown for the *IPT2* of all the speed factors. (**C**) Pairwise comparisons of the *IPT2* are represented in terms of the speed factor within *DH* and (**D**) *NDH*. (**E**) Pairwise comparison of *IPT2* for both *DH* and *NDH*.

**Figure 5 sensors-20-03477-f005:**
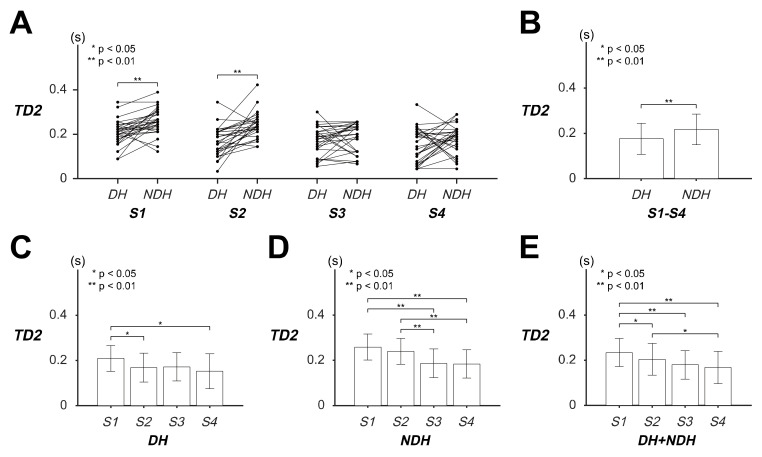
Evaluation of transient responses of the circular tracking movement based on the time delay (TD2) on the frontal plane. (**A**) The values of TD2 of 29 subjects at four speed factor levels. (**B**) The pairwise comparison between dominant hand (DH) and non-dominant hand (NDH) is shown for the TD2 of all speed factors. (**C**) Pairwise comparisons of TD2 are represented for each speed factor with the DH and (**D**) NDH. (**E**) Pairwise comparison of TD2 with both the DH and NDH.

**Table 1 sensors-20-03477-t001:** Characteristics of 29 Subjects.

Subject No.	Age (Years)	Sex	Dominant Hand
**1**	25	M	R
**2**	24	M	R
**3**	24	F	L
**4**	24	M	R
**5**	25	M	R
**6**	22	F	R
**7**	22	F	R
**8**	22	F	R
**9**	22	F	R
**10**	24	M	L
**11**	26	M	R
**12**	22	M	R
**13**	22	F	R
**14**	23	F	R
**15**	23	F	R
**16**	22	F	R
**17**	35	M	R
**18**	24	F	R
**19**	26	M	R
**20**	25	M	R
**21**	23	M	R
**22**	26	M	R
**23**	26	M	R
**24**	24	M	R
**25**	23	M	R
**26**	23	M	L
**27**	25	M	R
**28**	33	M	R
**29**	30	M	R

**Table 2 sensors-20-03477-t002:** Statistical results of each parameter.

Variable	Hand Factor	Speed Factor				Total
		*S1*	*S2*	*S3*	*S4*	
***IPV2*** **(%)**	*DH*	216.01 (±40.992)	204.53 (±36.119)	192.56 (±42.834)	178.31 (±24.868)	197.85 (±39.011)
	*NDH*	235.65 (±44.350)	215.90 (±38.679)	180.60 (±20.895)	172.86 (±25.152)	201.25 (±42.034)
	*DH + NDH*	**225.83** **(±43.471)**	**210.21** **(±37.532)**	**186.58** **(±33.944)**	**175.58** **(±24.942)**	
***IPT2*** **(s)**	*DH*	0.4299 (±0.1142)	0.4789 (±0.1126)	0.4954 (±0.1126)	0.4540 (±0.1105)	**0.4646** **(±0.1138)**
	*NDH*	0.4966 (±0.1513)	0.4966 (±0.1306)	0.5084 (±0.1087)	0.5261 (±0.1837)	**0.5069** **(±0.1448)**
	*DH + NDH*	0.4632 (±0.1371)	0.4877 (±0.1212)	0.5019 (±0.1099)	0.4900 (±0.1546)	
***TD2*** **(s)**	*DH*	0.2092 (±0.0571)	0.1686 (±0.0645)	0.1720 (±0.0626)	0.1525 (±0.0768)	**0.1756** **(±0.0681)**
	*NDH*	0.2586 (±0.0579)	0.2395 (±0.0571)	0.1870 (±0.0637)	0.1843 (±0.0626)	**0.2173** **(±0.0679)**
	*DH + NDH*	**0.2339** **(±0.0622)**	**0.2040** **(±0.0701)**	**0.1795** **(±0.0631)**	**0.1684** **(±0.0713)**	

* These values are recorded in terms of ‘mean (± standard deviation)’.

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
