# Peer review of "Analysis of Control Characteristics between Dominant and Non-Dominant Hands by Transient Responses of Circular Tracking Movements in 3D Virtual Reality Space"

_sensors, 2020, doi:10.3390/s20123477_

Round 1
Reviewer 1 Report
The manuscript titled "Analysis of control characteristics between dominant and non-dominant hands by transient responses of circular tracking movements in 3D virtual reality space" examined the control characteristics of dominant and non-dominant hands using a circular tracking movement in 3D virtual reality space.
The authors have presented a very well-written manuscript. The introduction provides sufficient background detail and is organized in an efficient format. The conclusions are well supported by the data, which have been thoroughly detailed in the results section.
Methodology
- were participants provided with a familiarization session to become acquainted with the HTC Vive HMD and interacting with the 3D virtual reality environment?
- were participants asked about their experience level using 3D virtual reality applications? if there were beginners and advanced users in the participant pool, could this affect the results?
- was there consideration to randomize the order of speed (i.e. S1, S2, S3, S4) for each participant? could participants have improved their performance from S1 to S4 based on becoming more efficient at the task through learning?
Clinical Application and Future Work
- this section provides an applied context to the current work. However, expanding on the clinical application would be beneficial. The current section discusses how future work could be conducted using a clinical population. Expanding on the concept of how the current data could be applied to the clinical population and potentially improve rehabilitation programs or tracking progress throughout a rehabilitation program would provide additional value to the manuscript.
Reviewer 2 Report
The authors present an experimental study, based upon a virtual reality system, for analysing the effect of three factors (target speed, dominant-non dominant hand, the interaction between the hand and speed factors) on three parameters (maximum velocity, time of maximum velocity, reaction time).
Comments:
1) Is it correct to write "visually guided target"?
2)Figure 1B: the unit of measure mm is not fully visible
3) Line 130: "They were also informed that four “beep” sounds would be played to indicate a 3-second countdown. " Does it mean that the subject was asked to start after the 4th beep? Please, clarify
4) Line 139 (and other parts of paper): "The target speeds were set at 0.125, 0.250, 0.500, 139 and 0.750 Hz along the trajectory. " You are speaking about angular velocity and, according to the International system of units, it is measured in rad per second (rad*s-1). Herz is the measure of frequency. So, if you write about target speed you should report the measure in rad/s
5) Line 147 "Cartesian (X,Y,Z) data measured at 90 Hz from the tracer were used for data analysis.". Please, improve this sentence. I think you want to say that the sampling frequency is 90 Hz
6) Line 149 "The performance indicated how the subject initiated target-tracking using the tracer after the countdown." I suggest to improve this sentence. It explains the aim of this paper
7) Line 161: "The 160 terms vx,trg [t] and vy,trg [t] were calculated as vx,trc [t] and vy,trc [t]. vxy,trg was considered as a constant". Why do you need to compute vx,trc and vy,trc if you have vxy,trc as a constant?
8) Lines 164-173: Please, improve the description of how you measure TD2. You can explain it better. You can clarify that subjects show "tremor" before the start and you definition allow to discriminate between erroneous involuntary movements from the voluntary one.
If I am not in error, TD2 is generally named reaction time and there are different definitions for it (from control theory to human movement science). I suggest you to evaluate the other definitions for the future works.
9) Introduce equations 8-12. Even if I understood, you didn't introduce -270, init etc..
10) Line 249 "0.0423 seconds faster than the NDH, ". Please, check. Maybe it is correct to say "before than"
11) Line 253 "0.0423 seconds less time than ". Please, check.
12) Improve the description of the analysis you perform on data. Clarify what are the aims and how you want to reach them.
13) I suggest to report some information about the trajectory performed by the subjects. Were they able to keep the circular trajectory? We know that there is a tradeoff between speed and accuracy in movements. I suggest the authors to read the following papers and to consider them for their future works or as reference to be included:
-Plamondon, R., & Alimi, A. M. (1997). Speed/accuracy trade-offs in target-directed movements. Behavioral and brain sciences, 20(2), 279-303.
-Parziale, A., Senatore, R., & Marcelli, A. (2020). Exploring speed–accuracy tradeoff in reaching movements: a neurocomputational model. Neural Computing and Applications, 1-27.
14) One of the conclusions of the paper is that "This also suggests that an identical internal model gives motor commands to the hands. ". This result is already known as "motor equivalence". I suggest you to read the following papers (other can be found the the two papers i cited above):
-Wing, A. M. (2000). Motor control: Mechanisms of motor equivalence in handwriting. Current biology, 10(6), R245-R248.
15) I suggest to improve the discussions and the conclusions. I know that this kind of system can be useful for further invenstigations but the results presents in this paper need to be better promoted.
